# Non-Conventional Metal Ion Cofactor Requirement of Dinoflagellate Alkaline Phosphatase and Translational Regulation by Phosphorus Limitation

**DOI:** 10.3390/microorganisms7080232

**Published:** 2019-08-01

**Authors:** Xin Lin, Chentao Guo, Ling Li, Tangcheng Li, Senjie Lin

**Affiliations:** 1State Key Laboratory of Marine Environmental Science (MEL), Xiamen City Key Laboratory of Urban Sea Ecological Conservation and Restoration (USER), Xiamen University, Xiamen 361102, China; 2Department of Marine Sciences, University of Connecticut, Groton, CT 06405, USA

**Keywords:** dinoflagellate, alkaline phosphatase, metal cofactor, inducible expression

## Abstract

Alkaline phosphatase (AP) enables marine phytoplankton to utilize dissolved organic phosphorus (DOP) when dissolved inorganic phosphate (DIP) is depleted in the ocean. Dinoflagellate AP (Dino-AP) represents a newly classified atypical type of AP, PhoA^aty^. Despite While being a conventional AP, PhoA^EC^ is known to recruit Zn^2+^ and Mg^2+^ in the active center, and the cofactors required by PhoA^aty^ have been contended and remain unclear. In this study, we investigated the metal ion requirement of AP in five dinoflagellate species. After AP activity was eliminated by using EDTA to chelate metal ions, the enzymatic activity could be recovered by the supplementation of Ca^2+^, Mg^2+^ and Mn^2+^ in all cases but not by that of Zn^2+^. Furthermore, the same analysis conducted on the purified recombinant ACAAP (AP of *Amphidinium carterae*) verified that the enzyme could be activated by Ca^2+^, Mg^2+^, and Mn^2+^ but not Zn^2+^. We further developed an antiserum against ACAAP, and a western blot analysis using this antibody showed a remarkable up-regulation of ACAAP under a phosphate limitation, consistent with elevated AP activity. The unconventional metal cofactor requirement of Dino-AP may be an adaptation to trace metal limitations in the ocean, which warrants further research to understand the niche differentiation between dinoflagellates and other phytoplankton that use Zn–Mg AP in utilizing DOP.

## 1. Introduction

In marine ecosystems, the availability of P is critical to the growth of phytoplankton. Thus, P is often found to be the limiting nutrient of the primary productivity in the ocean [1,2]. In the ocean, P is present in both inorganic and organic forms, among which dissolved inorganic phosphorus (DIP) as the preferred form for phytoplankton is quickly consumed and found at a low concentration in the euphotic zone [3]. In contrast, dissolved organic phosphorus (DOP) is relatively abundant in the euphotic zone. DOP mainly occurs in two forms: The major form phosphoesters and the less abundant form phosphonates [4,5]. When facing the DIP limited condition, phytoplankton are able to utilize a broad spectrum of phosphoesters to meet their cellular P requirement and sustain their primary production [6,7].

Enzymatic hydrolysis by alkaline phosphatase (AP) is a well-known mechanism employed by marine phytoplankton to utilize various types of phosphoesters which commonly features a phosphoester (O–P) bond. AP can release Pi from phosphoesters at the characteristic alkaline pH in the ocean [8]. Therefore, the presence and measurement of the bulk AP activity of phytoplankton has been used to indicate the P limitation in the ocean [2,7]. To gain further insights into the DOP utilization conveyed by AP in individual taxa, many studies have been conducted to examine the expression of AP coding genes and to characterize the enzyme properties in both marine bacteria and eukaryotic phytoplankton [9,10,11,12]. Experimental and genomic studies have revealed multiple types of APs which show not only substrate preferences but also different required metal ions as cofactors [13,14,15,16,17].

The conventional AP (PhoA^EC^) initially identified in *Escherichia coli* was the best studied and defined as a homodimeric phosphomonoesterase. It mainly hydrolyzes phosphomonoesters with low activity on the phosphate diester and requires 2Zn–Mg as bimetallo activity cores in the protein structure [18,19]. PhoA^EC^ has been documented in marine bacteria, cyanobacteria and diatoms [12,13,20]. The second type of AP with a determined protein folding structure is PhoX, which possesses a unique 3Ca–2Fe core [17] and exhibits a low substrate specificity for DOPs with an O–P bond [21,22]. Besides marine prokaryotes, PhoX homologs have also been identified in green algae, *Volvox carteri* and *Chlamydomonas reinhardtii* [23,24,25]. The third, rather common, type of AP in marine prokaryotes is PhoD, which has also been confirmed to be Ca^2+^-dependent and able to hydrolyze both phosphomonoesters and phosphodiesters [13,26].

In addition, there are several uncategorized types of APs in marine phytoplankton. A Ca^2+^-dependent putative AP was identified from the pelagophyte *Aureoumbra lagunensis* through a proteomic analysis [27]. A novel kind of AP, EHAP1, has also been identified in haptophyte *Emiliania huxleyi* [28]. Besides possessing an up-regulated PhoD under a P limitation [29], the diatom model *Phaeodactylum tricornutum* has been reported to own another type of AP with some sequence similarity to PhoA^EC^, but its enzyme activity is enhanced by Mn^2+^, Mg^2+^ and Ca^2+^ instead of Zn^2+^ [12,30]. Because of their high sequence variability, the structural and functional classification of these APs remains challenging. Though PhoA^EC^ has been documented in marine bacteria, it is less frequently reported than the other two types in marine microbes. Taken together, the multiple types of APs and the prevalence of Ca^2+^-dependent APs are believed to be an adaptive mechanism which gives different marine microorganisms their niche advantages to utilize available types of phosphoesters under low metal ion availability conditions in the ocean [22].

Dinoflagellates are one of the most important groups of phytoplankton in the ocean and are well known to be capable of utilizing DOP with the aid of an AP under DIP-depleted conditions. Reports showed that the subcellular localization of the Dinoflagellate-AP (Dino-AP) can be both cell surface and intracellular [11,31]. Based on the genomic and transcriptomic data of the several dinoflagellate species currently available, only one type of AP has been identified in dinoflagellates, and we classified it as the PhoA^aty^ type due to a generally high sequence divergence but recognizable conserved domains similar to an atypical PhoA type in cyanobacteria [32]. In these PhoA^aty^ APs, we found conserved aspartic acid (D), glutamic acid (E) and a “proline-aspartic” motif amongst those conserved domains, a motif known to be responsible for Ca^2+^-binding, thus suggesting that PhoA^aty^ might also be a Ca^2+^-dependent type of AP [32]. However, this hypothesis has not yet been examined experimentally.

In this study, we investigated the metal ion requirement of APs in five dinoflagellate species (*Amphidinium carterae*, *Alexandrium pacificum*, *Karenia mikimotoi*, *Prorocentrum minimum* and *Fugacium kawagutii*) using an EDTA chelating and metal ion resupply approach. To better demonstrate the AP dependency on the metal ion, we overexpressed the AP gene of *A. carterae* (ACAAP) in *E. coli*, and enzymatic assays conducted on purified recombinant ACAAP (*r*ACAAP) verified that this AP could be activated by Ca^2+^, Mg^2+^, and Mn^2+^ but not Zn^2+^. We further developed a polyclonal antiserum against ACAAP, and a western blot analysis showed the remarkable up-regulation of ACAAP expression under a phosphate limitation and a positive correlation between ACAAP and AP activity, confirming the active function and translational regulation of this gene.

## 2. Materials and Methods

### 2.1. Algal Strains, Culture Conditions and Sample Collection

Five dinoflagellate strains were used in this study. *A*. *carterae* and *F*. *kawagutii* (formerly *Symbiodinium kawagutii*) [33] were provided by the Provasoli-Guillard National Center for Marine Algae and Microbiota (NCMA); *A. pacificum* (formerly *A*. *catenella*) and *P*. *minimum* were provided by Center for Collections of Marine Bacteria and Phytoplankton, Xiamen University (CCMBP); and *K*. *mikimotoi* was provided by Jinan University. These strains were maintained in sterilized oceanic seawater (filtered through 0.22 µm pore size filters, salinity 30 psu) enriched with the full nutrient regime of the L1 medium with no Si [34] in an incubator with temperature controlled at 20 °C (most examined strains) or 25 °C (*F. kawagutii*) and illumination under a light dark cycle L:D = 12:12 (most examined strains) or 14:10 (*F. kawagutii*) with a photon flux of 100 µE m^−2^ s^−1^.

DIP-depleted and replete cultures (under the same condition as described above) were prepared in triplicates. The DIP-depleted cultures were grown under the same nutrient regime as DIP-replete cultures, except for a decreased phosphate concentration (2 µM). The cell density in the culture was determined daily by the counting of the cells using a Sedgewick–Rafter counting chamber (Phycotech, St. Joseph, MI, USA). The DIP concentration in the culture media was measured using the Molybdate Blue method [35]. Cells from each culture (200 mL) were harvested by filtration onto 3 μm pore size polycarbonate membranes (Millipore, Bedford, MA, USA) and washed off the membrane using an AP buffer (20 mM Tris-HCl, pH 8.0, 25 mM NaCl). Harvested cells were homogenized by bead beating three times separately, using 0.5 mm ceramic beads with the setting of 6 m s^−1^ for 30 s on an MP FastPrep-24 (MP Biomedicals, CA, USA). After centrifugation at 13,000 *g* at 4 °C for 10 min, the supernatant that contained extracted crude protein was subjected to enzyme activity measurements.

### 2.2. Alkaline Phosphatase Activity Measurement on Algal Cells and Protein Extract

The bulk AP activity of cultures was measured by adding 50 μL of 20 mM p-nitro-phenylphosphate (pNPP from Fluka, St. Louis, MO, USA), prepared in 1 M Tris-HCl (pH 8.0) into a 1 mL culture sample [36]. The reaction was carried out in sterile microcentrifuge tubes and incubated in darkness for 2 h at 25 °C. After centrifugation at 10,000 *g* for 2 min, the supernatant was removed and used for an OD measurement at 405 nm on a NanoDrop ND-2000 spectrophotometer (Thermo Scientific, Wilmington, DE, USA).

An in-gel AP activity assay was performed as described previously [27] with minor modifications. The extracted protein of 10^6^
*A*. *carterae* cells of both the P-depleted and the P-replete groups were incubated with a non-reducing Laemmli sample buffer at room temperature for 5 min and loaded into (4%–10%) SDS-PAGE gel, with an equal amount of protein in each well (25 μg) and in triplicate for the P-depleted group. After electrophoresis, each gel lane was excised, soaked in an AP buffer (20 mM Tris-HCl, pH 8.0, 25 mM NaCl) containing one of the three different metal ions (10 mM Ca^2+^, 10 mM Mg^2+^, 0.1 mM Zn^2+^), and incubated with ELF97 (1:50 dilution) (ELF^TM^ 97 endogenous phosphatase detection kit, Molecular Probes, OR, USA) for 2 h in the darkness at room temperature. For the P-replete group, each gel slice was subjected to the incubation with all three metal ions included to detect possible AP activity in one (instead of multiple) assay. Fluorescent gel images were acquired on the Bio-Rad Gel Doc XR system using an EtBr filter at 302 nm (Bio-Rad, CA, USA).

### 2.3. Heterologous Expression of Recombinant ACAAP and Production of Antibody

Based on the AP coding gene (*acaap*, GenBank: HQ259111.2) identified in *A*. *carterae* [36], two different kinds of amplicons were prepared and subjected to heterologous expression. A hydrophilic peptide (*p*ACAAP) comprising 180 amino acid residues (nucleotide site 661–1200) was used to generate a polyclonal antibody, and a recombinant ACAAP (*r*ACAAP) comprising 684 amino acid residues (full ORF region with the exclusion of a signal peptide coding region (nucleotide site 1–60)) was used for an enzyme activity analysis. Both gene fragments were amplified using specific primers (Table 1), and then gel-purified PCR products were cloned into vector pEasy-E1 (TransGen Biotech, Beijing, China) and transformed into *E*. *coli* BL21 (DE3). For each amplicon, a single colony containing the gene fragment was grown separately in a 3 mL LB (Luria-Bertani) liquid medium containing 100 µg mL^−1^ ampicillin on a shaker rotating at 200 rpm at 37 °C for 16 h; each colony was then transferred to 1 L fresh LB with 0.5 mM IPTG (isopropyl-β-D-thiogalactopyranoside) to induce the expression of the inserted gene fragment, grown for 3 h at 37 °C for *p*ACAAP and 10 h at 28 °C for *r*ACAAP, both on a shaker rotating at 200 rpm.

*E. coli* cells were harvested by centrifugation at 5000 *g*, 4 °C for 10 min, resuspended in a 1 mL Tris-HCl buffer (50 mM Tris-HCl, pH 8.0), and then homogenized by ultrasonic treatment. After centrifugation at 15,000 *g*, 4 °C for 10min, the supernatant containing the heterologously-expressed peptide was purified using an Ni-NTA spin kit column (TransGen Biotech, Beijing, China) following the manufacturer’s protocol. The resultant overexpressed peptide was loaded into a Superdex 75 gel filtration column on an AKTA prime liquid chromatography system (GE Healthcare Bio-Sciences, Uppsala, Sweden) and eluted using a phosphate-buffered saline buffer (PBS buffer, 50 mM NaH_2_PO_4_, 150 mM NaCl, pH 8.0) as the mobile phase. All collected fractions were examined by sodium dodecyl sulfate polyacrylamide gel electrophoresis (SDS-PAGE), and fractions containing the target peptide were concentrated using Amicon Ultra centrifugal filter devices (Merck Millipore Ltd., Carrigtwohill, IRL). Purified *r*ACAAP was subjected to further enzymatic characterization, and purified *p*ACAAP was used to immunize a rabbit (Japanese white) to raise a polyclonal antiserum (Proteintech Group Inc., which has the laboratory animal license SYXK2014-0076 issued by Hubei Science Technology Department, Wuhan, China).

### 2.4. Western Blot Analysis

Crude protein, extracted as described above, was mixed with a reducing Laemmli buffer and incubated at 95 °C for 5 min. The denatured proteins of each sample were loaded into (4%–10%) SDS-PAGE gel (Bio-Rad, CA, USA) at equal amounts in each well. Electrophoresis was carried out at 90 V for 30 min, followed by 120 V for 60 min. The SDS-PAGE gel was then blotted onto a polyvinylidene fluoride (PVDF) membrane (Millipore, Bedford, MA, USA) at 25 V for 30 min using the Trans-Blot SD semi-dry transfer cell (Bio-Rad, CA, USA). The membranes were incubated in a blocking buffer containing 5% non-fat milk for 2 h at room temperature, followed by incubation with the polyclonal antiserum (1:5000 dilution) in TBS (Tris buffered saline) for 1 h at room temperature. After four washes with TBS containing 0.1% Tween 20 (TBST), the membranes were incubated with a biotinylated goat anti-rabbit IgG (TransGen Biotech, Beijing, China) in a 1:10,000 dilution for 1 h at room temperature and then were washed four times in TBST. Meanwhile, the reference protein GAPDH (glyceraldehyde-3-phosphate dehydrogenase) was also examined using the monoclonal antibody (1:5000 dilution, Sangon Biotech, Shanghai, China) in parallel following the same procedure. The immunodetected bands were visualized using the enhanced chemiluminescent (ECL) substrate (Bio-Rad, CA, USA), and chemiluminescene images were acquired on a Bio-Rad Gel Doc XR system (Bio-Rad, CA, USA). The relative quantification of ACAAP expression was calculated in two ways, normalized to the ACAAP expression of P-replete on day 1 and normalized to reference protein GAPDH based on the band intensity estimated by Quantity One software (Bio-Rad, Hercules, CA, USA).

### 2.5. Metal Ion Dependency Analysis

The metal ion requirement of APs was investigated in five dinoflagellate species and the purified *r*ACAAP. The five dinoflagellate species were grown under the DIP-depleted condition as described above, and cells were collected at the stationary phase. For each enzymatic reaction, an equal amount of proteins (equal to 10^4^~10^5^ cells dependent on different species) were used. Pre-incubation with EDTA was employed to chelate the cellular metal ions present in the protein. The reaction mixture contained 6 μg protein, 5 μL 100 mM EDTA, and 75 μL AP buffer, and it was incubated at 25 °C in the darkness for 30 min in a 96-well plate. Afterwards, 5 μL 20 mM pNPP was applied to the reaction mix along with 10 μL of one of each of the following: 100 mM Ca^2+^, 100 mM Mg^2+^, 100 mM Mn^2+^, 100 mM Co^2+^, 100 mM Fe^3+^, 10 mM Zn^2+^, and 100 mM EDTA (all prepared in 20 mM Tris-HCl pH 8.0), triplicate wells for each treatment were done separately [13]. After another 2 h incubation, AP activity (APA) was measured on a SpectraMax Paradigm (Molecular Devices, CA, USA), and the spectrum reading of the control treatment (Ctrl) was taken as a base value to normalize the readings of other treatments to calculate the relative fold change of the enzyme activity as follows: (APA–APA_Ctrl_)/APA_Ctrl_. Meanwhile, a control treatment was maintained in the preserved condition as pre-incubation, and 5 μL (30 U μL^−1^) commercialized CIAP (calf intestinal alkaline phosphatase, Takara Bio Inc., Kusatsu, Shiga, Japan) were subjected to the same procedure as a positive reference.

## 3. Results

### 3.1. Metal Ion Requirement of Dino-AP

The metal ion requirement analysis was performed on crude protein extracts of *A. carterae*, *K. mikimotoi*, *A. pacificum*, *P. minimum* and *F. kawagutii* collected under the P-depleted condition. After pre-incubation with EDTA, AP activity disappeared (Figure 1). The AP activity was recovered significantly with the supplement of Ca^2+^, Mg^2+^ and Mn^2+^ in *A. carterae*, *K. mikimotoi* and *A. pacificum* (Tukey HSD, *p* < 0.01, Figure 1a–c). In *P. minimum*, the AP activity was recovered significantly with the supplement of Mg^2+^ and Mn^2+^, while in *F. kawagutii*, the AP activity was recovered by Mn^2+^ but not significantly by Ca^2+^ and Mg^2+^ (Figure 1d,e). To exclude the possibility that the apparent lack of AP activation by Zn^2+^ was due to a toxic effect of overdose, the commercialized CIAP was used as a control in the assay. In sharp contrast to dinoflagellate APs, the enzymatic activity of the CIAP was significantly recovered by the supply of Mg^2+^ and Zn^2+^ (Tukey HSD, *p* < 0.01,) (Figure 1f), as documented of the conventional AP.

The same assay was tested on the purified *r*ACAAP (heterologously expressed ACAAP in *E. coli*) incubated with the six metal ions separately following the same method as for algal cellular protein extracts. The result was consistent with the observation of crude protein extracts of *A. carterae*, and the AP activity of rACAAP was recovered significantly with the supplement of Ca^2+^, Mg^2+^, and Mn^2+^ (Tukey HSD, *p* < 0.01), but not Zn^2+^, Fe^3+^ or Co^2+^ (Figure 1g). Meanwhile, the in-gel AP activity assay conducted on the crude protein extract of *A. carterae* revealed a unique positive protein band only in the P-depleted group with appended Ca^2+^, but there was negative in the P-depleted group with Mg^2+^ and Zn^2+^ and in the P-replete group (Figure 2).

### 3.2. Expression Pattern of AP in A. carterae (ACAAP) under Varied P Conditions

*A. carterae* was grown in normal phosphate (36 μM) and low phosphate (2 μM ) media. DIP concentration remained scarce (below the detection limit 0.2 μM) in the P-depleted group throughout the experiment. However, in the P-replete group, it was initially high (25 μM) and then decreased quickly, reaching as low as that in the P-depleted group after day four (Figure 3a). Accordingly, markedly different growth curves were observed between two groups of cultures (Figure 3b). During the first three days, the cell densities in both groups increased steadily comparably, from ~1.8 × 10^4^ to ~ 8.4 × 10^4^ cells mL^−1^. The cells in the DIP-replete group continued the exponential growth until day five and then entered the stationary growth phase in a stable cell density. At the end of this experiment, the cell density of the DIP-replete culture was 2.4 × 10^5^ cells mL^−1^, a 12-fold increase compared to the starting cell density. In the DIP-depleted group, the cultures grew more slowly after day three, and the cell density was 1.3 × 10^5^ cells mL^−1^ at the end of this experiment—only about half of that in the DIP-replete group.

By comparison, AP activity started to diverge between the two groups earlier than did the cell density (Figure 3c). At the beginning of the experiment, AP activity was barely detectable in both groups. On day three, when the cell density was about the same between the two groups, AP activity in the P-depleted group (196 fmol pNP cell^−1^ h^−1^) increased dramatically to five-fold of that in the P-replete group (37 fmol pNP cell^−1^ h^−1^). Since then, the AP activity in the P-depleted group continued to increase quickly, while that in the DIP-replete group only increased slightly despite the significant drop of DIP concentration observed after day three (Figure 3a). At the end of this experiment, the AP activity in the DIP-depleted group (430 fmol pNP cell^−1^ h^−1^) was about four-fold of that in the DIP-replete group (99 fmol pNP cell^−1^ h^−1^).

The extracted total proteins of *A*. *carterae* from both groups were subjected to SDS-PAGE and a western blot analysis. The result showed that the polyclonal antiserum raised against the *p*ACAAP specifically detected a unique band, which lied between 100 and 150 kDa (Figure 4a), apparently larger than computationally estimated 75 kDa of ACAAP.

The immunodetected protein exhibited a significant difference in abundance between P-replete and P-depleted groups. A clear band was detected on day two and became thicker in the course of the experiment, indicating the inducible expression of ACCAP (Figure 4a). A remarkable up-regulation in the expression of ACCAP was observed in the DIP-depleted group compared to that in the DIP-replete group, regardless which normalization method used to evaluate the fold change of expression level (T-test, *p* < 0.05) (Figure 4b,c). The expression of ACAAP in the P-depleted group reached the peak on day four and maintained at a stable level until day seven, which was about 80% of that on day four, 21-fold of that on day one, and 57-fold of that in the P-replete group on day seven (Figure 4b). In contrast, the expression of ACAAP in the DIP-replete group generally maintained at low levels (shown as faint detected bands in gel, Figure 4a) throughout the experiment, despite the slight increase after the cultures entered the stationary phase (Figure 4b,c). This result was consistent with the trend observed in AP activity (Figure 1c). Overall the protein expression of ACAAP was positively correlated with enzyme activity (R^2^ = 0.88, *n* = 10, Figure 4d).

## 4. Discussion

### 4.1. Inducible Expression of Dinoflagellate APs Through Transcriptional and Translational Regulation

AP activities have been shown to be induced by P limitations in various dinoflagellate species in previous work [36,37,38], with which the results of the current study are consistent. Previously, the induction has been attributed to the transcriptional up-regulation of the AP gene under P limitations in dinoflagellate *A*. *carterae*, *K*. *brevis* and *P*. *donghaiense* [36,37,39]. In this study, using the antiserum developed against ACAAP in a western blot analysis, we observed the marked stimulation of AP protein expression in the DIP-depleted *A*. *carterae*, which was positively correlated with the AP activity increase. However, the magnitude of the increase in ACAAP abundance was much higher than in its activity. All these results suggest that the regulation of ACAAP expression and enzymatic activity in response to phosphate limitation lies at both the transcriptional level and the translational level, which is in agreement with the expression observed in EH-PhoA^aty^ of *E. huxleyi* [40]. Furthermore, the increase of both AP activity and AP expression appeared earlier than growth arrest in the P-depleted culture of *A. carterae*, and we noted a slight increase of both in the stationary growth phase of the P-replete group when there was a sharp drop in DIP concentration. Our previous study showed that AP of *A*. *carterae* was localized at both cell surface and intracellular compartment [11]. Taken together, these observations suggest that *A*. *carterae* might employ AP to utilize the intracellular DOP storage to maintain population growth when both DIP and DOP were deficient in the medium. Therefore, the expression of AP in *A*. *carterae*, and likely in other dinoflagellates as well, can be triggered by a low ambient Pi concentration, but it is really regulated by the internal P availability, similar to the case of EH-PhoA^aty^ [40].

### 4.2. Evidence of Dinoflagellate AP Protein Modification from Observation on ACAAP

The generation of ACAAP antiserum has enabled us to estimate the molecular size, besides abundance, of a dinoflagellate AP based on a western blot analysis. We noticed that there was a discrepancy in molecular mass between the ACAAP detected in SDS-PAGE (~130 kDa) and the in silico prediction based on the amino acid sequence of ACAAP (~75 kDa). A striking coincidence occurred in PhoA^aty^ of *E*. *huxleyi* (EH-PhoA^aty^), in which while the amino acid sequence gave a predicted molecular mass of 75 kDa, the detected band in the western blot lied between 100 and 120 kDa, ~110kDa [40]. Our subsequent mass spectroscopic analysis showed that the ~110 kDa band was truly an EH- PhoA^aty^ protein. One possibility is that the mature PhoA^aty^ type AP, of which both ACAAP and EH-PhoA^aty^ are, may function in a dimer-hinge conformation like the typical AP, PhoA^EC^ [19]. Though protein complexes are usually broken in the protein denaturing buffer and SDS-PAGE, it cannot be ruled out that this AP holozyme may stay intact under these conditions. As documented, some stable dimers can remain un-dissociated in SDS-PAGE gel and able to perform enzyme activity [41,42]. Another possibility is that the post-translational modifications (PTMs) may occur and alter the molecular mass of the AP monomer or dimer, as has been reported previously for other proteins [43,44]. PTMs have been reported in algal APs. APs in the diatom *P*. *tricornutum* and the coccolithophorid *E*. *huxleyi* undergo proteolytic modification while released from the cell surface [12,28]. Our previous study proposed that the PhoA^aty^-type AP represents a phylogenetically distinct type of AP, sharing little sequence similarity to any other classified type of AP, and, as such, their substrate preferences and protein structure still remain undetermined [32]. PTMs are of great importance to the biological function of proteins, including the enzyme activity state, subcellular localization, turnover and interactions with other proteins [43]. Thus, further efforts are required to investigate possible PTMs of PhoA^aty^ and their effects on enzymatic function.

### 4.3. Metal Ion Requirement of Dino-AP

Metal ions are indispensable for many enzymes to form and stabilize the conformational structure of folded proteins, providing the specific function in substrates binding and hydrolysis [45,46]. Besides transition metals (Fe, Cu and Mn) previously found to be required particularly in biochemical redox process, the metalloproteins containing Mg, Ca and Zn have also been extensively studied [45]. For instance, the archetypal PhoA^EC^ has been well characterized and shown to require 2Zn–Mg in each subunit to coordinate the active center, while another major characterized type of AP, PhoX, owns a complex active-site comprising 2Fe–3Ca [17,19,46]. To implement the catalytic function, bimetallo cores require the transition metal Fe^3+^/Zn^2+^ to interact with the oxygen atoms of the O–P bond of the substrate molecule, the same as Ca^2+^ in PhoX [19,46]. Even though no available active site and structural model could be applied to PhoA^aty^, we have identified conserved motifs in PhoA^aty^, including residues of aspartic acid (D) and glutamic acid (E), that feature a high affinity to Ca^2+^ [47] and a highly conserved Ca^2+^-binding “proline-aspartic acid (PD)” motif shared by PhoX and phosphotriesterase [17,21,32,48].

Consistent with the prediction, experiments in this study showed that the AP activity eliminated by metal iron depletion could be restored by the resupply of Ca^2+^, Mn^2+^ or Mg^2+^ in all five examined dinoflagellate strains but not by that of Zn^2+^. It is worth noting that *E*. *huxleyi* was able to express two different types of AP (EH-PhoA^aty^ and EHAP1) at the same time, and the AP activity of cell lysate (excluding most EHAP1 because it is secretory) could be restored more by Ca^2+^ than Zn^2+^ [40]. This result is comparable to the previous study carried out in the AP of other algae and PhoX of bacteria. Besides Ca^2+^ and Fe^3+^, the activity of PhoX can also be recovered with the presence of Mn^2+^ (shown in supplementary data in [17]) and Mg^2+^ [48]. A cell-surface AP isolated from *P*. *minimum* also exhibited the activity with the presence of both Ca^2+^ and Mg^2+^ [49,50], and the AP in dinoflagellate *Pyrocystis noctiluca* has previously been described as Mg^2+^-dependent [51]. A similar cofactor requirement was also observed in the diatom *P*. *tricornutum*, in which PhoA activity could be enhanced by Mn^2+^, Mg^2+^ and Ca^2+^, and the AP of the brown tide alga *A. lagunensis* was found to be Ca^2+^-dependent [12,27]. This compatibility may be due to the similar chemical characteristics shared by Ca^2+^ and Mg^2+^, as the Mg^2+^ binding residues also show preference for “D” as well as the Ca^2+^ binding site [52]. Nevertheless, there was a difference between the in-gel assay and the algal cell assay, in this study in that the resupply of Mg^2+^ and Mn^2+^ failed to recover AP activity (data not shown) in the in-gel assay, which can be explained as SDS-PAGE may disable protein conformational changes needed to accommodate these bivalent cations [53].

The prevalence of the Ca^2+^-dependent AP reported in marine microorganisms has been attributed to the relative low availability of Zn^2+^ in the ocean and the high abundance of Ca^2+^ and Mg^2+^ in the ocean [1,14,16,22,27]. Therefore, the adoption of Ca^2+^ in an AP may be an adaptive strategy that has evolved in the marine microorganisms in the Zn–P co-limited environment. However, the identification of the Ca^2+^/Fe^3+^ active site of PhoX poses a possible Fe-deficiency based constrain of AP activity and a nutrient limitation-driven environmental selection of microorganism in the ocean [17,54]. We also noticed that the activity of the Dino-AP could also be recovered by the addition of Mn^2+^, one common transition metal found in metalloproteins, which, like Ca^2+^, is more abundant in the sea than Zn^2+^ [1].

In the PhoA^EC^ family, the structural flexibility of the bimetallic site has been examined to interpret the promiscuous catalytic activity [46]. Structural comparisons of AP superfamily members showed distinct structure features outside of the conserved bimetallo site, which may not only impact specific substrate binding but may also modulate the properties of the bimetallo core to recognize substrates [19]. However, it is still unclear how Ca^2+^ and Mn^2+^ function in the active center of Dino-AP. Our results showed negative effect of Zn^2+^ on AP activity recovery in dinoflagellate, we also found resupply of Zn^2+^+Mg^2+^, like resupply of Zn^2+^ alone, did not recover AP (data not shown). However, without a further high-resolution structural observation of the AP proteins with metal ions inside, caution is required to interpret it as the dispensability or even antagonism of Zn^2+^ for dinoflagellate APs. Though we do not have crystallographic data to directly show the metal ion conformation of the Dino-AP protein, the possibility of an experimental artifact—for instance the addition of Zn^2+^, albeit only 1% dosage of other metal ions, caused toxic effects to the Dino-AP in our assays—is unequivocally excluded by the result of exactly the same assay for the calf intestinal alkaline phosphatase (CIAP), which exhibited total recovery of AP activity by the resupply of Zn^2+^+Mg^2+^, as expected of a conventional AP. 

## 5. Conclusions

Dinoflagellate AP belongs to a newly classified type of AP, PhoA^aty^, and its enzyme expression and characterization has remained largely unexplored. This study is the first attempt to examine the metal ion cofactor requirement of the AP in core dinoflagellate species and develop an antibody to quantify AP abundance. Our results from five dinoflagellate species revealed that Dino-AP was activated by Ca^2+^, Mg^2+^ and Mn^2+^ but not by Zn^2+^. Additionally, the AP activity could be restored to different degrees by above-mentioned divalent cation among different species, in contrast to the conventional AP which requires the pair of Zn^2+^+Mg^2+^ as cofactors. Furthermore, our results showed for the first time that the mature enzyme of dinoflagellate AP^aty^ might function in a dimer-hinge conformation or undergo a post-translational modification that alters the molecular mass dramatically. These results, albeit a bit surprising, are not entirely unprecedented considering the similar behavior of the PhoA^aty^ AP (EH-PhoA^aty^) in the haptophyte *E*. *huxelyi* and of other proteins in other organisms. Therefore, we propose here, with caution, that PhoA^aty^ may require unconventional metal ions such as Ca^2+^ as cofactors, which is likely a result of adaptive evolution under the selection pressure of limited or unpredictable trace metal availability in the ocean. Further study is required to illuminate the protein structure of PhoA^aty^ and to determine the exact cofactor conformation in order to gain further understanding of the regulation of growth and the distribution of dinoflagellates from the perspective of P nutrient strategy in the trace metal variable marine environment.

## Figures and Tables

**Figure 1 microorganisms-07-00232-f001:**
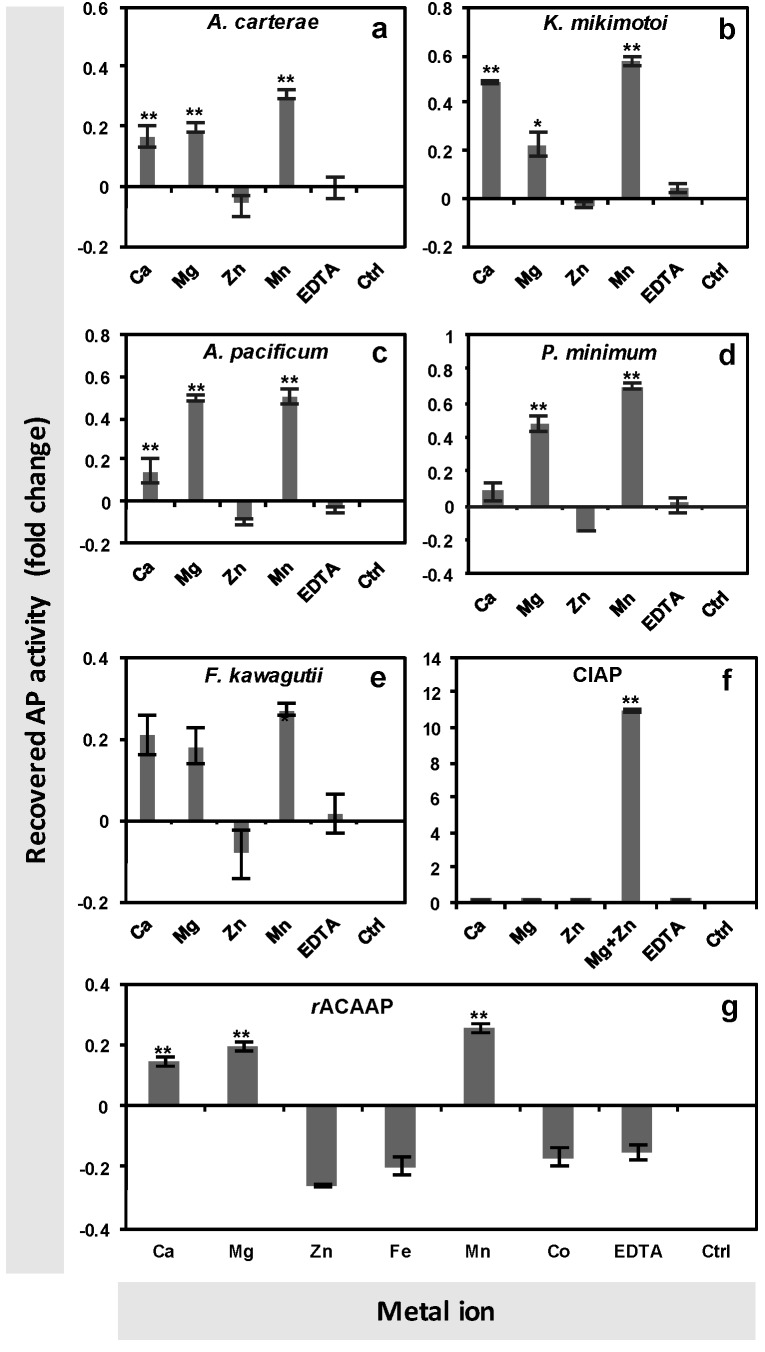
Alkaline phosphatase activity (APA) recovery of crude proteins of different dinoflagellate strains ((**a**) *A*. *carterae*, (**b**) *K*. *mikimotoi*, (**c**) *A*. *pacificum*, (**d**) *P*. *minimum*, (**e**) *F*. *kawagutii*), the commercialized CIAP (**f**) and the recombinant AP of *A. carterae* (*r*ACAAP) (**g**) with the resupply of different metal ions after EDTA treatment to remove metals and eliminate APA. “Ctrl” represents the EDTA-pre-incubated reaction, which served as a blank control here. “EDTA” represents an experimental control that was resupplied with the same molar amount of EDTA as metal ions in experimental reactions. Asterisks depict the significant recovery of APA by metal ion resupply compared to the EDTA control (Tukey HSD, ** represents *p* < 0.01, * represents *p* < 0.05).

**Figure 2 microorganisms-07-00232-f002:**
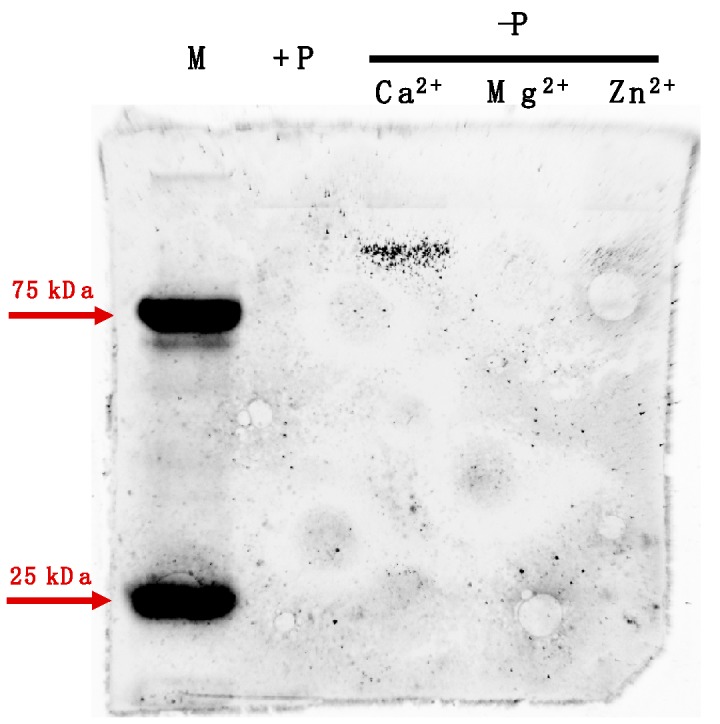
In-gel assay of metal ion specificity required by AP of *A. carterae*. “M” represents protein marker, “+P” represents extracted protein of the P-replete group, “−P” represents extracted protein of the P-depleted group incubated with specific metal ion respectively.

**Figure 3 microorganisms-07-00232-f003:**
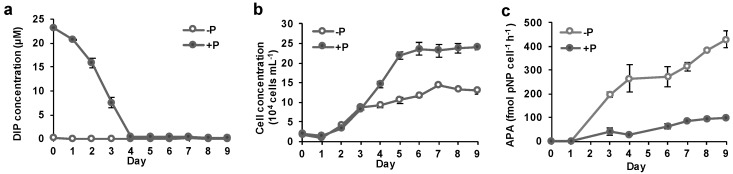
Dissolved inorganic phosphate (DIP) concentration (**a**), growth curve (**b**) and AP activity (**c**) in time course of *A*. *carterae* cultures under different phosphate conditions. +P, normal phosphate concentration in an L1 medium; −P, decreased phosphate concentration (2 μM) in an otherwise L1 medium.

**Figure 4 microorganisms-07-00232-f004:**
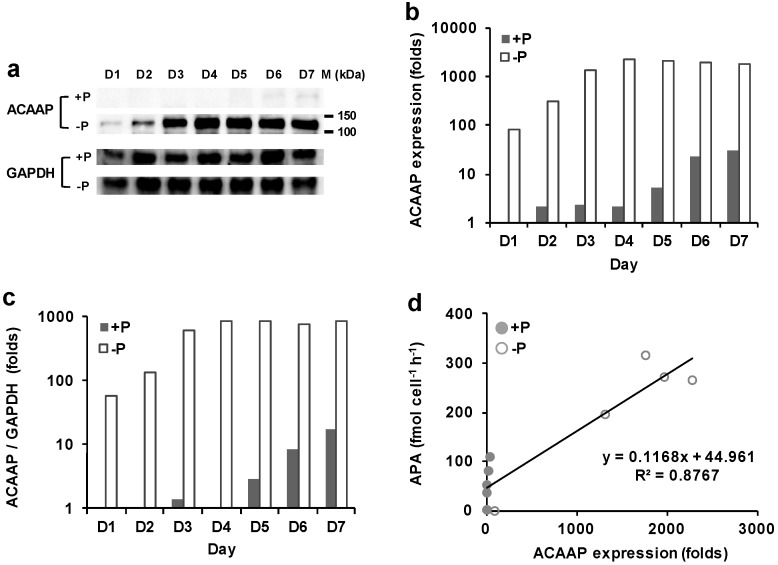
AP expression in *A*. *carterae* cultures grown under DIP-replete (+P) and depleted (−P) conditions. (**a**) ACAAP and glyceraldehyde-3-phosphate dehydrogenase (GAPDH) western blot. (**b**) Relatively quantified ACAAP expression. (**c**) ACAAP expression normalized to GAPDH. (**d**) Pearson correlation between ACAAP expression and APA (*p* < 0.05).

**Table 1 microorganisms-07-00232-t001:** Primers used in this study.

Applications	Primer Name	Direction	Primer sequence (5′-3′)	PCR Condition
*p*ACAAP	PF1	F	TTGAGCACATACACCGACGA	94 °C 3min, 35 cycles (94 °C, 30 s, 56 °C, 45s, 72 °C, 1 min), 72 °C, 10 min
	PR1	R	AGATGCATTCAGATACATGATG	
*r*ACAAP	PF2	F	AAGGGACGTAGGCTTGCTAG	
	PR2	R	CGCACGCACGGTCAAGAAG	

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
