# Peer review of "Non-Conventional Metal Ion Cofactor Requirement of Dinoflagellate Alkaline Phosphatase and Translational Regulation by Phosphorus Limitation"

_microorganisms, 2019, doi:10.3390/microorganisms7080232_

Round 1
Reviewer 1 Report
The manuscript attempted to carry out basic characterization of recombinant proteins of alkaline phosphatase.
AP is a family of important enzymes
However, there are serious problems without sequence definition of AP types, which were not defined in ms; for that matter, how the attempted experiments relate to other orthologues/homologues is not possible.
The experimental design was inundated with major problems for addressing enzyme activities,
Protein and antibodies characterization were completely lacking, especially if molecular weight were interpreted as “dramatic” post-translational modification (predicted size 75 to 130kD !!) which could well be the background of the anti-serum against dinoflagellate cell lysates.
Were the cell lysis protocol consistent with the location this specific dinoflagellate AP ?? or was most of the enzymes depleted in the first clearing spin after cell lysis ??
Optimum temperatures, ?? defined minimal medium which might have affected the apparent “metal dependency “
None these crucial points were even considered in the ms.
The English problems extended throughout the ms.
Reviewer 2 Report
In the manuscript, “Non-conventional metal ion cofactor requirement of dinoflagellate alkaline phosphatase and translational regulation of phosphorus limitation” Xin Lin et al. present data suggesting the impact of depleted inorganic phosphorus on alkaline phosphatase expression from dinoflagellates. The authors also provide data concerning the functional metals able to coordinate within the active site of these atypical APs. The presented work is fairly thorough, and the methods used were typical and well described. However, the manuscript requires significant editing prior to publication to improve the text’s ability to appropriately convey the authors’ results. Below are some concerns and specific edits for the authors.
Following these changes, the covered material is appropriate for publication in microorganisms.
---
1. Page 2; lines 50-51 and 64-66: The authors initially state that PhoAEC has been documented in marine bacteria (lines 50-51), and then later state (lines 64-66) that PhoAEC are rarely reported from marine bacteria…this seems a bit confusing as written and should be clarified.
2. Page 4; lines 157-158: The authors do not include any IUCUC or equivalent animal procedure approvals. I assume that these are required prior to publication.
3. Page 6; Figure 2: As presented, this figure is not very convincing and needs to be improved. Perhaps a higher resolution might benefit the authors’ presentation of this data.
4. Page 7; lines 265-266: How exactly did the authors generate the linear formula presented in Figure 4 that suggests a positive correlation? This is definitely not apparent as presented.
5. Page 8; lines 298-300: The authors’ justification that the AP dimer might stay intact during denaturing conditions requires some referencing. Especially considering they cite this result as evidence of an unproven dimer-hinge conformation in their conclusions. As presented, I do not consider the authors’ data to support conclusion that these Aps function in a dimer-hinge conformation.
Other suggested edits:
Page 1; line 30: “In marine ecosystem” ecosystem should be plural and the referencing isn’t formatted correctly.
Page 1; lines 36-38: The final sentence of paragraph 1 requires heavy edits for clarity.
Page 1; line 39: “the best known mechanism” reword this
Page 1; lines 40-41: “form O-P bond at the characteristic alkaline pH in the ocean” this should be edited for clarity
Page 1; lines 43-44: “many studies have been reported to identify and characterize AP” this requires editing for clarity.
Page 2; lines 45-46: as written, it reads as though the metal ions can also be substrates
Page 2; lines 65-67: the final sentence of paragraph 2 is difficult to parse as written
Page 2; line 75: The final sentence of paragraph 3 should be rewritten to better convey the authors’ intention here
Page 2; lines 78-79: the sentence beginning with “To better demonstrate” requires rewording.
Page 3; line 99: the phrase “Daily cell concentrations” is a bit awkward…I’m not really sure concentration is the appropriate word here.
Page 3; lines 127-128: The first sentence of section 2.3 is unclear as written.
Page 3; line 134: The phrase “single colony of each gene fragment” is really unclear and requires editing.
Page 4; line 148: “the expression peptide” should be reworded
Page 5; line 201: organism name should be in italics
Page 6; line 214: “heterogeneously” should be changed to heterologously
Page 6; lines 231-233: This sentence should be reworded for clarity.
Page 7; lines 250-252: The authors should also reconsider this sentence.
Page 7; line 274: “as well as current study” is a bit abrupt and unclear.
Page 8; lines 285-286: “luxurious uptake and store up” is not really working as written and should be changed
Page 8; line 325: “re-activated” is a bit misleading here. Maybe “restored” would work better here.
Page 9; lines 356-357: “Despite the direct proof is yet to come” should be reworded for clarity.
Throughout text: inconsistent spacing between numbers and scientific units throughout text; inconsistent use of the Oxford comma when presenting items in a series.
Reviewer 3 Report
p.p1 {margin: 0.0px 0.0px 0.0px 0.0px; font: 12.0px Helvetica} p.p2 {margin: 0.0px 0.0px 0.0px 0.0px; font: 12.0px Helvetica; min-height: 14.0px}Lin and co-authors present a well-executed and insightful study on the biochemistry of alkaline phosphatase in dinoflagellates. The authors highlight an atypical PhoA alkaline phosphatase from Amphidinium carterae (ACAAP) and describe its induction under low DIP availability, post translational modification, and unusual metal co-factor requirement. In addition, cell lysates from a number of other dinoflagellate species were examined, with results depicting a similar profile of metal dependence as the recombinant ACAAP (for example, the lack of dependence on Zn). Overall, results support the broad prevalence among dinoflagellates of an atypical alkaline phosphatase and its unconventional metal cofactor requirement, suggesting possible adaptation to trace metal limitation in the ocean by this key group of phytoplankton. While I found the manuscript to be scientifically strong and well written overall, I have several points for the authors to consider.
It is said that the activity of metal-depleted dinoflagellate APA is not restored with additions of Zn, while in contrast, typical PhoA is dependent on Zn. However, I find this to be misleading. The activity of typical PhoA (CIAP in Figure 1f) is only restored with the addition of Zn + Mg and not Zn alone. Similarly, Zn alone did not restore APA in dinoflagellate samples, yet results from Zn + other metals were not presented for dinoflagellates. So the possibility remains that dinoflagellate APA could be dependent on Zn in the presence of other metals such as Mg. Was this tested?
It would be helpful to see some context about other potential AP’s present in dinoflagellate genomes. Do dinoflagellates possess putative homologs of typical PhoA (PhoAEC), PhoX, and/or PhoD? If not, then PhoAaty may indeed contribute quantitatively to overall AP activity in these organisms. If they do possess other AP’s, then the significance of PhoAaty to the P nutritional metabolism of these microorganisms may be somewhat less. Either way, some background information on this point seems pertinent, since some of the results are from bulk lysates, which could potentially include a diversity of APs found in the genome.
Line 40 and elsewhere - what does O-P bond refer to? The P-monoester or diester bond classes? Phosphoanhydride bond class? Or all of these? I think the authors mean the P-monoester bond class (P-O-C), but this is unclear. Please clarify.
Figure 2 and Figure 1g seem to present a contradictory result. I could not find any discussion of why the microplate assay showed dependence of rACAAP activity on Mg but the in-gel activity assay showed a lack of dependence on Mg. Could the authors provide any insight on this discrepancy?
Line 99, line 225 and elsewhere, the term “reduced phosphate” is confusing and could be mistaken to indicate P in its reduced oxidation state such as +3.
For the data in figure 1, please indicate which growth condition (+P or -P) and in which growth phase the cell lysates were prepared, as these details bear significantly on P nutritional status.
Line 285 - I suggest replacing “luxurious” with “luxury”
Sentence ending on line 330 - add a parenthesis at the end of the sentence.
